# How does moving public engagement with research online change audience diversity? Comparing inclusion indicators for 2019 & 2020 European Researchers' night events

**Aaron M. Jensen**[1], **Eric A. Jensen**[1], **Edward Duca**[2], **Jennifer Daly**[3], **Niamh Mundow**[4], **Joseph Roche**[3]*

**1** Institute for Methods Innovation, Dublin, Ireland, **2** University of Malta, Msida, Malta, **3** Trinity College Dublin, Dublin, Ireland, **4** University College Cork, Cork, Ireland

* Joseph.Roche@tcd.ie

**Data Availability Statement:** The anonymized dataset underlying the results presented in the article are available on the Zenodo database at DOI:

## Abstract

Taking place annually in more than 400 cities, European Researchers' Night is a pan-European synchronized event that aims to bring researchers closer to the public. In this paper audience profiles are compared from events in 2019 and 2020. In 2019, face-to-face events reached an estimated 1.6 million attendees, while in 2020, events shifted online due to the COVID-19 pandemic and reached an estimated 2.3 million attendees. Focusing on social inclusion metrics, survey data is analyzed across two national contexts (Ireland and Malta) in 2019 (n = 656) and 2020 (n = 506). The results from this exploratory, descriptive study shed light on how moving public engagement with research online shifted audience profiles. Based on prior research about the digital divide in access and use of online media, hypotheses were proposed that online European Researchers' Night events would attract audiences with higher educational attainment levels and greater self-reported, subjective economic well-being. While changes were observed from 2019 to 2020, results for each hypothesis show a mixed picture. The first hypothesis was upheld for the highest education levels but failed for the lowest levels suggesting that the pivot to online events simultaneously attracted participants with no formal education and those with postgraduate qualifications, while attracting less of those with undergraduate or lower levels of education. The second hypothesis was not upheld, with online European Researchers' Night events attracting audiences with slightly higher levels of economic well-being compared to face-to-face events. The findings of this study indicate that European Researchers' Night events present a clear opportunity to measure the effects of the digital divide in relation to public engagement with research across Europe.

## Introduction

The tragedy of the COVID-19 pandemic created a unique 'natural experiment' scenario, allowing cross-year comparison between face-to-face (2019) and online (2020) iterations of the

10.5281/zenodo.5830027 (https://doi.org/10.5281/zenodo.5830027).

**Funding:** This work was supported by funding from the European Union's Horizon 2020 Research and Innovation Programme for several projects: PROBE (Grant Agreement no. 817914), START (Grant Agreement no. 955428), Cork Discovers (2019 Grant Agreement no. 818789, 2020 Grant Agreement no. 955330), Science in the City (2019 Grant Agreement no. 818730, 2020 Grant Agreement no. 955263), and QUEST (Grant Agreement no. 824634). Four authors received funding from the European Union's Horizon 2020 Research and Innovation Programme. ED received funding through grant no. 818730, JD received funding through grant no. 955428, NM received funding through grant nos. 818789 and 955330, and JR received funding through grant nos. 817914 and 824634. The funders had no role in study design, data collection and analysis, decision to publish, or preparation of the manuscript.

**Competing interests:** The authors have declared that no competing interests exist.

same public engagement events. This study explores data from such a scenario in two countries to compare differences in audience diversity for those who engage with and benefit from online and offline public engagement with research. The events compared were part of European Researchers' Night (ERN), an initiative of the European Union (EU) tasked with widening access and engagement with European research and innovation. As part of the EU's long-term mission, ERN has taken place annually across Europe since 2005 intending to showcase research, raising public awareness and interest, and strengthening the relationship between science and European society [1–3].

While comprising unique activities in every country, ERN broadly resembles science festivals, albeit focusing on a more expansive range of research disciplines. While science festivals have existed in some form since at least 1831 [4], they have dramatically increased in number and size in recent decades [5]. Such festivals are seen as celebrating scientific content and ideas to engage public audiences [6]. Studies that have explored why people attend science festivals have found visitors value direct interactions with researchers [7]. Similarly, ERN aims to ensure that European research is visible and participative in how research addresses pressing concerns that face European society.

This paper focuses on who engages with and benefits from research and innovation by exploring key indicators of social inclusion: educational attainment and self- reported, or subjective, household economic well-being status. Although a range of socioeconomic dimensions are not regularly measured among science festival audiences, studies have shown that visitors tend to be already comfortable in such environments due to being privileged in their educational attainment and socioeconomic status [8]. Likewise, studies of ERN have found high educational qualification levels among audiences [9,10]. Additionally, the digital divide discourse has consistently shown contributions, such as education and income, to inequalities in who accesses and benefits from the digital landscape [11–14].

Combined with a unique context created by the COVID-19 pandemic, this study compares differences in audience profiles for 2019 (offline) and 2020 (online) iterations of ERN events. Given the literature on the digital divide, we hypothesized that the 2020 online ERN events would attract an audience skewed towards higher levels of educational attainment and those with greater levels of subjective economic well-being.

## European researchers' night: Background and changing context

European Researchers' Night is a flagship initiative of the European Union tasked with widening access and engagement with European research and innovation. The European Commission provides guidance that ERNs should bring researchers closer to the general public and suggests that a combination of education and entertainment will be most effective to engage with people regardless of the level of their scientific backgrounds [15,16]. With an emphasis on younger audiences, the European Commission suggests a range of activities including hands-on experiments, science shows, simulations, debates, games, competitions, and quizzes. Many event organizers include arts-themed activities such as performances, theatre, stand-up comedy, short stories, and art installations [9,10,17]. These events were free of charge to all attendees in both 2019 and 2020.

The COVID-19 pandemic created a 'natural experiment' scenario, allowing cross- year comparison between public engagement events that were predominantly face-to-face in 2019 and then shifted online in 2020 [18]. In 2019, ERN events were organized over two days at the end of September and ran face-to-face events in over 400 cities across Europe, with over 1.6 million attendees reported [19]. In 2020, due to the COVID-19 pandemic, ERNs were moved to the end of November. Event coordinators were allowed to spread their activities over a

longer period and encouraged to shift events online, with over 2.3 million attendees reported [19]. The overall approach to digital marketing adopted by the three events evaluated here was broadly similar in 2019 and 2020. This empirical study explores shifts in audiences' profiles from well-established ERN events in two national contexts in 2019 and 2020. Malta and Ireland were selected as the focus for this study because coordinators of these ERN events agreed to collaborate on this research.

First, we briefly introduce the events studied below:

## European researchers' night in Malta (2019 & 2020)

In Malta in 2019, European Researchers' Night was reported as the largest national science and arts festival, attracting around 30,000 people. The 2019 iteration of the festival had an emphasis on face-to-face activities. It took over a substantial area of the historic capital of Valletta by running interactive performances, art installations and hands-on activities in its streets and buildings. In the 2020 iteration, the festival went online for the first time by running several digital pre-festival events over a whole month of November and then culminated in a 'marathon live streaming video' displayed on a range of digital platforms, including Zoom and Facebook Live. The 2020 iteration was estimated to engage more than 20,000 people in total with online activities that included a STEM escape room, performances, science shows, puppet theatre, pre-recorded theatre and video content, debates, quizzes, question and answer sessions and other formats. Both 2019 and 2020 iterations had marketing campaigns that reached an estimated 300,000 people.

## European researchers' night in Ireland (2019 & 2020)

In Ireland, European Researchers' Night includes Cork and Dublin, as different cities with data collected independently over multiple years. In 2019 iterations, ERN activities primarily took place on university campuses over the last weekend of September, including lecture theaters, labs and other campus spaces that are usually occupied by academic staff and students. Activities included walk-up events and allowed people to participate in tours, discussions, viewing posters and science-themed arts and demonstrations. In the 2020 iterations, all ERN activities were shifted online for the first time over the last weekend of November.

The Cork ERN events in 2019 primarily took place on the campus of University College Cork. Public audiences moved between exhibition and information stands to interact with researchers, watch demonstrations, and participate in experiments. In 2020, the event formats changed entirely from face-to-face to virtual. Extensive preparation was required in the lead up to the last weekend of November, with two live-streamed TV shows organized and led by workshops with researchers. These researchers trained in creating short, pre-recorded videos to give an overview of their research projects. These videos were edited and proofread, uploaded to the Cork ERN website, and a selection of researchers were chosen to appear in the live-streamed shows. The Friday night live show targeted adults, and the Saturday morning was dedicated to younger audiences. In addition to these live shows, scientific experiments that could be carried out at home, such as creating yogurt, were advertised to families. Researchers at Teagasc, an Irish research agency in the agri-food sector, organized an experiment box for interested schools in the same week of the live events and researchers carried out Zoom calls with primary schools.

The Dublin ERN events in 2019 were held on the university campus at Trinity College Dublin. The 2019 iteration took the form of a free, public pop-up festival to highlight the diverse range of academic research in Dublin. Held in Trinity's historic Front Square in Dublin city center, attendees could participate in live experiments, exclusive performances, interactive

workshops, stand-up comedy, and storytelling sessions. The organizer for the Dublin event changed between 2019 and 2020 iterations. While the events took place on the same university campus, adjustments were introduced in branding and advertising, including a change to the event's name. Organizers of the 2020 iteration aimed to disperse events around the campus and Dublin city. However, the Irish government placed a strict lockdown in Dublin and organizers had to switch to an entirely virtual event because no in-person activities were permitted. The shift to online delivery required extensive preparation to revise the program in the weeks leading up to the last weekend of November. The revised program included 60 separate live virtual sessions for 27 different activities and a parallel program of "any time" activities for people of all ages. Additionally, organizers developed a program for primary school children consisting of live science workshops and history tours with researchers providing live interactive virtual sessions in classrooms around Ireland to approximately 1,500 schoolchildren.

### Moving public engagement with research online

In this study, each of the established ERN events changed their program of activities from face-to-face in 2019 to online, virtual events in 2020. This shift was driven by the COVID-19 pandemic, mandated restrictions and concerns for health and safety. Within this unique context, education and many basic services were shifted online. This shift has highlighted longstanding gaps in digital infrastructures, such as the availability and access to broadband internet, interdependencies on digital devices, digital literacy, and inequalities in who can access and benefit from the digital landscape [11–14]. Several studies indicate key socioeconomic factors, such as education and income, contribute to the digital divide via digital and material access [12,13]. In this manuscript, we build on relevant studies regarding the internet and its effects on social inclusion [3], namely access and engagement with European research and innovation.The organizers of the 2020 iteration of ERN moved events primarily online for the first time, using a range of digital platforms to fulfil their mission to showcase research and widen access and engagement with European research and innovation. This shift of public engagement with research online can be seen as testing the 'rosy scenario', whereby the internet can "level the playing field and strengthen the voice of the voiceless" [11]. However, disadvantaged groups within society may experience barriers following the shift towards greater reliance on digital and material access to online platforms [11]. We focus on the contribution of education and socioeconomic well-being as two factors repeatedly identified in the digital divide literature [1–5].

### Research hypotheses

By exploring indicators of social inclusion among those who engage with and benefit from research [20], and given the high educational qualification levels among ERN audiences [9,10,21,22], coupled with the digital inequities [11–14] caused by the COVID-19 pandemic, the following hypotheses were formulated:

**Hypothesis 1 (H1):** Online ERN events in 2020 will attract an audience with *higher educational attainment* compared to face-to-face events in 2019.

**Hypothesis 2 (H2):** Online ERN events in 2020 will attract an audience with *greater levels of subjective economic well-being* compared to face-to-face events in 2019.

## Methods

This research received ethical approval from an ethics committee at Trinity College Dublin. The ensuing sections describe the methods and procedures used to gather audience survey responses and the subsequent analysis. This study utilized secondary data—survey respondents provided electronic, written consent at the events for their anonymized data to be used

for research purposes in academic publications. The research approach balanced the practical compromises needed for real-world naturalistic exploratory research and sampling best practices, such as having equal probability of selection as well as random allocation to treatment and control groups. This is often difficult in audience research settings, where public audiences have free choice about where they spend their leisure time [23,24].

## Instrument

The survey instrument was administered in English in both 2019 and 2020 iterations of ERN events. The questionnaire used closed-ended multiple-choice questions (e.g. demographic data and Likert scales about attitudes towards research). The questions analyzed for this paper focus primarily on quantitative nominal and ordinal data (e.g. educational attainment and household income) to compare to broader population data across multiple countries. Educational attainment was measured at the individual level with the following question: "What is the highest level of education you have completed?". Response options in the 2019 iteration included more categories for educational attainment below an undergraduate degree than 2020 and were combined post-hoc to align both years. Economic well-being [25] was measured at the household level using a self-report scale, with the following question: "Please indicate what your household can usually afford". Response options were focused on different abilities to meet basic needs.

## Procedure

Procedures for data collection in 2019 and 2020 event iterations had some similarities and differences. In 2019, all event sites used similar data collection protocols. For example, on the day of the event, adult attendees were approached by data collection volunteers and asked if they were willing to provide answers to a few questions on-site and then respond to a follow-up questionnaire sent by email after the event. The 2019 iteration used a systematic on-site 'intercept' sampling approach to mimic random selection to the extent feasible when the audience research setting is face-to-face events [10,23].

The most substantial differences in data collection procedure for the 2020 iterations were based on greater reliance for digital-only research methods and integrations between different digital technologies, including public-facing event websites, booking platforms (such as Eventbrite) and a digital research platform for GDPR-compliant data collection capabilities provided by Qualia Analytics (qualiaanalytics.org). Differences were observed in how each event interfaced with the research platform and invited participation. Still, all sites used a similar method after respondents were invited in pre-event and follow- up questionnaires. Events not using the booking platform used the digital research platform to streamline enrollment in the research.

## Data analysis

Unweighted data were used in the analysis. Some total percentages presented in the results add to less than 100 due to either rounding of decimals, exclusion of response categories or questions that have multiple response options. The analysis was completed using Qualia Analytics' (qualiaanalytics.org) built-in dashboard, along with SPSS 27 and Microsoft Excel. The significance threshold for all tests was $\alpha = .05$. National population statistics were utilized in this study as a basis for comparison to address the hypotheses [26–28].

## Sample

We analyzed audience profiles for those attending European Researchers' Nights events held in 3 cities across Ireland and Malta in 2019 and 2020. This analysis produced descriptive

**Table 1. Sample frame distribution of ERN survey respondents by country.**

| Country | Invited | | Responded | |
|---|---|---|---|---|
| | n = | % | n = | % |
| Malta | 1601 | 65 | 667 | 57 |
| Ireland | 877 | 35 | 495 | 43 |
| **Total** | **2478** | **100** | **1162** | **100** |

statistics about the audience profiles across each country, year of attendance, and event cases. For all events, the total sample frame of invited respondents (N = 2478) was dispersed between Malta (65%, n = 1601) and Ireland (35%, n = 877). The specific levels of achieved sample size are indicated based on available data for Malta (57%, n = 667) and Ireland (43%, n = 495). The response rate comparison for attendees by country is presented in Table 1.

The data collection was conducted across multiple years. The total sample frame of respondents was distributed between 2019 (29%, n = 722) and 2020 (71%, n = 1756). The specific levels of achieved sample size are indicated based on available response data for 2019 (57%, n = 656) and 2020 (43%, n = 506). The sample frame distribution by year of data collection is presented in Table 2.

While a much larger sample frame is indicated in 2020, the survey was carried out differently in each year of data collection. In 2019, the survey was conducted on the day of each event through face-to-face intercept data collection at entrances to the events. For 2019 events, the total sample frame of invited respondents was determined by those willing to participate (78%, n = 566;) and those who declined to participate (12%, n = 156) at the point of intercept. Subsequently, all events in 2020 were moved online and the survey was carried out digitally with either self-enrollment available on event websites or automated enrollment and email invitations that were connected to booking systems and dependent on the technical capacity of each event organizer.

The data collection presents multiple event cases conducted across each year and location. In Ireland, Case 1 and Case 2 share the same institution and city, but the event organizers changed between 2019 and 2020, and so are presented as unique cases. Whereas, Case 3, also in Ireland, and Case 4, in Malta, share the same event organizers across both years. The comparison for data collection by country, year and event cases is presented in Table 3.

## Results

This study assessed audience profiles in terms of demographic diversity and representativeness of the wider public. Here, we begin with levels of educational qualification as a key indicator of social inclusion across 2019 and 2020.

### Hypothesis 1: Educational attainment

Considering the aggregate across all countries, cases and event locations, most respondents in 2019 indicated having at least some university-level education (65%, n = 404), with most

**Table 2. Sample frame distribution by year of data collection.**

| Year | Invited | | Responded | |
|---|---|---|---|---|
| | n = | % | n = | % |
| 2019 | 722 | 29 | 656 | 56 |
| 2020 | 1756 | 71 | 506 | 44 |
| **Total** | **2478** | **100** | **1162** | **100** |

**Table 3. Comparison of data collection by country, year and case.**

| Country | Case | Year Collected | | Invited | | | | | | Responded | | | | | |
|---|---|---|---|---|---|---|---|---|---|---|---|---|---|---|---|
| | | 2019 | 2020 | 2019 | | 2020 | | Total | | 2019 | | 2020 | | Total | |
| | | 2019 | 2020 | n = | % | n = | % | n = | % | n = | % | n = | % | n = | % |
| Ireland | Case 1 | Yes | No | 110 | 15% | – | – | 110 | 4% | 109 | 17% | – | – | 109 | 9% |
| | Case 2 | No | Yes | – | – | 514 | 29% | 514 | 21% | – | – | 140 | 28% | 140 | 12% |
| | Case 3 | Yes | Yes | 114 | 16% | 139 | 8% | 253 | 10% | 114 | 17% | 132 | 26% | 246 | 21% |
| Malta | Case 4 | Yes | Yes | 498 | 69% | 1103 | 63% | 1601 | 65% | 433 | 66% | 234 | 46% | 667 | 57% |
| | | | | 722 | 100% | 1756 | 100% | 2478 | 100% | 656 | 100% | 506 | 100% | 1162 | 100% |

holding degrees at undergraduate (32%, n = 196) or postgraduate (33%, n = 208) levels. Whereas a much lower portion of respondents indicated having below undergraduate (35%, n = 217) and no formal qualification (0%, n = 1). Similarly in 2020, most respondents indicated having at least some university-level education (73%, n = 404), with most holding degrees at undergraduate (29%, n = 141) or postgraduate (44%, n = 212) levels. Again, a lower portion of respondents indicated having below undergraduate (18%, n = 86) and no formal qualification (9%, n = 45). The comparison for educational attainment of respondents across all events in 2019 and 2020 is presented in Table 4.

These results show observable changes from 2019 to 2020. For example, the two categories with an increase are 'no formal qualification' (+9%) and 'postgraduate' (+11%) levels, while the two categories with a decrease are 'below undergraduate' (-17%) and 'undergraduate' (-3%) levels. The observed ±8% shift between 'below undergraduate' (2019, 35% ⇒ 2020, 27%) and 'above undergraduate' (2019, 65% ⇒ 2020, 73%) categories show that ERN attendees are consistently highly educated [10].

Furthermore, we found that a larger proportion of adult attendees to European Researchers' Night events were more highly educated compared to the respective national populations (See Table 5). Compared with national figures for each country, these figures have shown a consistent *overrepresentation* of university-educated attendees (or degree holders). Indeed, as a combined segment, degree holders attending ERN events in Malta and Ireland represented more than 50% above the respective national populations in both 2019 and 2020. However, across both countries a smaller portion of attendees in 2019 and 2020 held 'undergraduate degrees' (Malta: +26% ⇒ +20%; Ireland: +17% ⇒ +13%) while a larger portion held 'postgraduate degrees' (Malta: +24% ⇒ +32%; Ireland: +33% ⇒+41%) than the respective national populations. The comparison for the educational attainment of respondents across Malta and Ireland in 2019 and 2020 compared to national populations is presented in Fig 1.

Compared with national figures for each country, these figures have shown a consistent *underrepresentation* of below university-educated attendees (or non-degree holders). As a combined segment, non-degree holders attending ERN events in Ireland represented 29%

**Table 4. Comparison of educational attainment of respondents across all event locations in 2019 & 2020.**

| | 2019 | | 2020 | |
|---|---|---|---|---|
| | n | % | n | % |
| No formal qualification | 1 | 0% | 45 | 9% |
| Below undergraduate degree | 217 | 35% | 86 | 18% |
| Undergraduate degree (Bachelor's or equivalent) | 196 | 32% | 141 | 29% |
| Postgraduate degree (Master's, PhD or equivalent) | 208 | 33% | 212 | 44% |
| **Total** | **622** | **100%** | **484** | **100%** |

**Table 5. Comparison of educational attainment of the national population and participating ERN audiences in 2019 & 2020.**

| | | National Population | ERN Sample Comparisons | | | | | | |
| | | | 2019 | | | 2020 | | | Diff |
| Qualifications | Country | % | % | | +/- % | % | | +/- % | +/- % |
|---|---|---|---|---|---|---|---|---|---|
| No qualification | Malta | 48 | 0 | | -48 | 16 | | -32 | +16 |
| | Ireland | 8 | 0 | | -8 | 4 | | -4 | +4 |
| Below Undergraduate degree | Malta | 42 | 40 | | -2 | 23 | | -19 | -18 |
| | Ireland | 44 | 23 | | -21 | 14 | | -30 | -9 |
| Undergraduate degree | Malta | 6 | 32 | | +26 | 26 | | +20 | -6 |
| | Ireland | 18 | 35 | | +17 | 31 | | +13 | -4 |
| Postgraduate degree | Malta | 3 | 27 | | +24 | 35 | | +32 | +8 |
| | Ireland | 10 | 43 | | +33 | 51 | | +41 | +9 |

below the respective national populations in 2019 and 34% below the respective national populations in 2020. Whereas non-degree holders attending ERN events in Malta represented 49% below the respective national populations in 2019 and 51% below the respective national populations in 2020. By comparison, the proportion of ERN attendees with qualifications 'below undergraduate degree' level decreased from 2019 to 2020 (Malta: -1%; ⇒ -19%; Ireland: -21% ⇒ -30%). However, the percentage of attendees with 'no qualification' went up (Malta: -48% ⇒ -32%; Ireland: -8% ⇒ -4%) in 2020.

ERN attendees in the sample across all years and countries have shown underrepresentation of those with 'no qualification' compared with national population statistics. However, Malta's results are particularly striking for this category, with an increase (16%) in participation in this category from 2019 to 2020. Nevertheless, there is still a substantial underrepresentation of those with no qualifications in Malta, given that almost half (48%) of its population has no formal education qualification.

Considering the relationships in the sample for this study, we conducted a chi-square test of independence comparing the 2019 and 2020 Qualification Levels. A significant interaction was found with a large effect size (DF = 3, $X^2$ = 91.951, p < .001, Cramér's V = 0.288), indicating a strong relationship between Qualification and Year that predicts 8.29% of the variance.

## Hypothesis 2: Subjective economic well-being

We note that between 73–82% of ERN respondents in each country and year indicated an ability of their household to afford 'All needs or more' (Malta: 78.7% ⇒ 80.1%; Ireland: 73.7% ⇒ 81.4%). In contrast, a much lower portion of respondents indicated an ability of their household to afford 'Some needs but not all' (Malta: 21.3% ⇒ 19.9%; Ireland: 26.3% ⇒ 18.6%). Within these categories, we observed overall shifts of ±1.4% in Malta and ±7.7% in Ireland

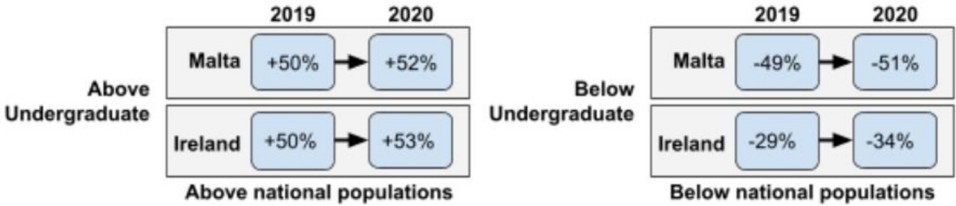

**Fig 1. Comparison of educational attainment of the national population and participating ERN audiences in 2019 & 2020.**

**Table 6. Response rate comparison of subjective economic well-being across ERN events in 2019 & 2020.**

| | | ERN Sample Comparisons | | |
| | | 2019 | 2020 | Diff |
| Subjective Indicators | Country | % | % | +/- % |
|---|---|---|---|---|
| Some needs but not all | Malta | 21.3 | 19.9 | -1.4 |
| | Ireland | 26.3 | 18.6 | -7.8 |
| All needs or more | Malta | 78.7 | 80.1 | +1.4 |
| | Ireland | 73.7 | 81.4 | +7.7 |

between years. The comparison for the subjective economic well-being of respondents across each country in 2019 and 2020 is presented in Table 6.

We conducted a chi-square test of independence comparing years and subjective economic well-being in both countries. This test found no significant interaction between Economic Well-being and Year (DF = 1, $X^2$ = 0.474, p = .491).

## Discussion

In this paper, the profiles of audiences engaging with online and offline public engagement with research events were compared, taking advantage of the natural experiment conditions created by the pandemic. This research combined evaluation evidence from two European countries and several cities in a cross-year comparison between face-to-face (2019) and online (2020) iterations of the same public engagement events. All these events were part of European Researchers' Night (ERN), an initiative of the European Union (EU) aimed at widening access and engagement with research.

Shifts in demographic indicators of social inclusion were identified between the same set of 2019 (offline) and 2020 (online) events. Given the literature on the digital divide, it was hypothesized that the 2020 online engagement events would draw an audience skewed towards higher levels of educational attainment and those with greater levels of subjective economic well-being. The results do not support this hypothesis. Instead, it was found that audiences' level of educational attainment was more polarized in the online context: There were increased prevalence at both the lowest and highest ends of the attainment spectrum. In other words, the online events enabled greater participation of both the highly educated and those with no educational qualifications. These findings suggest the possibility that moving public engagement with research online may have had a democratizing effect on participation.

Also based on the digital divide literature, it was hypothesized that the online events in 2020 would garner an audience with greater levels of subjective economic well-being compared to the face-to-face events in 2019. The results weakly supported the hypothesis, with already high levels of subjective economic well-being reported by audiences in the offline events repeated in the online events. The audiences in Ireland were somewhat more likely to be in the highest subjective economic well-being category in the online context, but the effect size was extremely small for this variable overall.

Ultimately, the results indicate that the effects of shifting public engagement events online are mixed when it comes to social inclusion. On one of the key dimensions tested, there was an increase in audience diversity (participants with no educational qualifications). While on most dimensions, the indicators were stable (remaining skewed towards high socio-economic status) or decreasing in social inclusion. Therefore, the findings demonstrate that it is possible to leverage the online engagement pathway to enhance social inclusion, despite the headwinds presented by the digital divide. Yet, there remains a steep hill to climb for public engagement with research to achieve more equitable audience participation.

## Author Contributions

**Conceptualization:** Aaron M. Jensen, Eric A. Jensen.

**Data curation:** Aaron M. Jensen, Eric A. Jensen.

**Formal analysis:** Aaron M. Jensen, Eric A. Jensen.

**Funding acquisition:** Edward Duca, Jennifer Daly, Niamh Mundow, Joseph Roche.

**Investigation:** Aaron M. Jensen, Eric A. Jensen.

**Methodology:** Aaron M. Jensen, Eric A. Jensen.

**Project administration:** Edward Duca, Jennifer Daly, Niamh Mundow, Joseph Roche.

**Software:** Aaron M. Jensen.

**Supervision:** Eric A. Jensen.

**Writing – original draft:** Aaron M. Jensen, Eric A. Jensen, Edward Duca, Jennifer Daly, Niamh Mundow, Joseph Roche.

**Writing – review & editing:** Aaron M. Jensen, Eric A. Jensen, Edward Duca, Jennifer Daly, Niamh Mundow, Joseph Roche.

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
