## [Decision Letter · Decision Letter 0]

27 Oct 2021

PONE-D-21-27097How Does Moving Public Engagement with Research Online Affect Audience Diversity? Comparing Inclusion Indicators for 2019 & 2020 European Researchers’ Night EventsPLOS ONE

A Dhuine Uasail Dr. Roche,

Thank you for submitting your manuscript to PLOS ONE. After careful consideration, we feel that it has merit but does not fully meet PLOS ONE’s publication criteria as it currently stands. Therefore, we invite you to submit a revised version of the manuscript that addresses the points raised during the review process.

 Please see the more detailed responses below. The responses were wide ranging and it's important to acknowledge the views of all the reviewers and respond to each, of which I look forward to receiving.

We look forward to receiving your revised manuscript.

Le gach dea-ghu,

Dylan A Mordaunt, MB ChB, FRACP, FAIDH

Academic Editor

PLOS ONE

Journal Requirements:

2. Please amend your authorship list in your manuscript file to include author

Additional Editor Comments (if provided):

Thank you for your submission. We had widely ranging feedback, all of which I've included below. The article presents what some might call a case study and one of the authors has referred to as a natural experiment, each of which are used in different social sciences contexts, which was part of the challenge in editing this paper. With specific reference to the publication criteria (https://journals.plos.org/plosone/s/criteria-for-publication):

1. The study appears to present the results of original research.

2. Results do not appear to have been published elsewhere.

3. Experiments, statistics, and other analyses are require some elaboration and further detail, as outlined by the reviewers below.

4. Conclusions are presented in an appropriate fashion and are supported by the data.

5. The article is presented in an intelligible fashion and is written in standard English.

6. The research meets all applicable standards for the ethics of experimentation and research integrity.

7. The article adheres to appropriate reporting guidelines and community standards for data availability.

Reviewers' comments:

Reviewer's Responses to Questions

**Comments to the Author**

1. Is the manuscript technically sound, and do the data support the conclusions?

Reviewer #1: Yes

Reviewer #2: Yes

Reviewer #3: Yes

Reviewer #4: Yes

2. Has the statistical analysis been performed appropriately and rigorously? 

Reviewer #1: No

Reviewer #2: Yes

Reviewer #3: Yes

Reviewer #4: I Don't Know

3. Have the authors made all data underlying the findings in their manuscript fully available?

Reviewer #1: No

Reviewer #2: Yes

Reviewer #3: No

Reviewer #4: Yes

4. Is the manuscript presented in an intelligible fashion and written in standard English?

Reviewer #1: Yes

Reviewer #2: Yes

Reviewer #3: Yes

Reviewer #4: Yes

5. Review Comments to the Author

Reviewer #1: Comment 1: This paper adopted a case, the 2019 & 2020 European Researchers’ Night Events, to investigate the digital divide of online and offline participants, however, relying on survey methods. The limitations are obvious, conclusion and implications can only be understood in a narrow context, such as the European Researchers’ Night Events. Hence, from my perspective, a qualitative method seems more appropriate for this type of research.

Comment 2: The research questions including two hypotheses: audience participation in online events seem to have higher educational attainment and subjective economic well-being when compared to those engaged in offline activities, seems can not provide this field with enough contribution and implications. It is more like a course paper than a journal paper, and I recommend you rethink the RQs and Hypos based on what theoretical and practical implications you can provide to this field.

Comment 3: The sample method for this second hand survey data, which seems to have adopted a convenience sampling method, can not guarantee representativeness, which in turn undermines the conclusion and generality of your study.

Comment 4: In the data analysis part, more advanced statistical methods could be applied, instead of too descriptive evidence, to show direct and robust evidence for the reader’s understanding.

Reviewer #2: Summary

A descriptive review of the change in audience attending the ERN, an annual European science festival, as it transforms from physical to digital delivery in a pre and post-COVID-19 climate. The study compares two social inclusion metrics within nations Malta and Ireland, namely levels of education attained and self-reported economic well-being. The manuscript clearly delineates between the evening expo events and how online content has compensated with mixed media interactions available in the digital space. The author articulates their hypotheses, methods, and results understandably.

Hypotheses of increased educational attainment levels and greater self-reported economic well-being were both upheld and not. In the context of diversity, questions arise whether COVID-19 potentiated skews in education and or economic well-being could be explored further. The manuscript could be strengthened with other measures of diversity and additional examination into reasons for hypotheses not being upheld; if the data is available to support more investigation. A title change to remove diversity without any revision of the data would be less misleading.

Title: Diversity

Diversity and inclusion metrics could be disaggregated for gender or age as a key measure of social inclusion. Ethnicity and linguistics may or may not be relevant or easy to compare given the two nations lack of direct comparison; however, the survey was entirely completed in English. Differences between events, and those speaking English as a first or second language at both events may reveal other influences on audience participation, socio-economic status and or other barriers to accessing the event eg. disability, isolation.

Title: Inclusion Indicators

Quantify the number of inclusion metrics as measures are not exhaustive. 2 inclusion indicators would frame the body of work better.

Abstract: Digital Divide

Income and education are often cited with age, linguistics, race and other factors in context of digital divide. Would help to acknowledge the communication infrastructure necessary to support digital delivery, comparison between the two locations Malta/Ireland and why other indicators have not been included in the statistical measure.

Introduction: Literature

Digital divide literature review methodology would be helpful to include.

ERN Night: Marketing campaigns

Aside from spend, were the demographic targeting settings for digital marketing the same for both? Could this have impacted the diversity of the audience / outcomes of the study?

Discussion: COVID-19

Given the pre and post-pandemic setting for the study, what relevance does it bring to bear on the digital divide and it’s impacts on diverse audiences’ engagement with science?

Reviewer #3: This study provides an interesting ‘natural experiment’ comparing in-person and online attendance at research engagement events pre- and mid-COVID pandemic (2019 vs 2020). This is a well-presented manuscript which will be of interest to many event organisers who continue to be impacted by COVID-19 in their ability to fully present in-person events and may have valid concerns about their ability to reach certain audiences online. The authors have done well to connect their results to the so-called ‘digital divide’, and with a few minor changes will be able to publish a study that has wide significance across numerous fields.

Minor comments:

The keywords provided are limited and repetitive: I recommend adding “science communication” or science/research engagement/outreach or “public engagement in science (or research)” or likewise to help researchers in those fields find this study when it is published.

Tables 1, 2 and 3 use the terms ‘enrolled’ and ‘achieved’ but these are not easy to understand out of context and do not appear well defined in text or in the table captions. Perhaps it would be cleared to use ‘recruited’ or ‘invited’ and ‘responded’, which would align more closely with the description in the methods.

From the in-text description it is clear which city ‘Case 1’ and ‘Case 2’ refers to, so I am unclear why they cannot simply be referred to by the city in the table, i.e. Dublin 1 and Dublin 2. It seems to introduce an unnecessary element of confusion to use different labels in the table when they are clearly identified elsewhere.

For the chi-square test of hypothesis 2, no significant effect has been detected and I think the current wording ‘an extremely weak statistical relationship’ is misleading. Please rephrase to make it clear no relationship was detected between economic wellbeing and year. However, you might consider running the test on Ireland alone to see if there is a difference when you separate country. Is there any economic difference between Ireland and Malta that would contribute to a different result?

In response to Q3 – have the authors made data full available: In the authors’ responses they have indicated data from the study will be available on request, whereas the PLOS Data Policy requires data to be fully available as part of the manuscript, supporting information or in a public repository. The authors should make their data available in such a manner, or explain why they are unable to meet the policy’s requirements.

Ethics statement – is there an approval number from the Trinity College ethics committee? This statement is quite vague: “an ethics committee” – is there more than one?

Major comments:

The introduction to the ERN events is generally very good and provides sufficient background to non-European readers. However, it might be useful to know if the European Commission has particular aspirations in terms of who is reached through these events, for instance those with less formal education or scientific qualifications. If such goals exist, it would be useful context in which to read this study’s findings and could be linked in the discussion in terms of whether a shift to online helped or hindered the reaching of these particular audiences.

It is not clear from the manuscript whether ERN events are free or ticketed. This is relevant information since moving events online would have a likely impact on cost (if there is any cost to attendees). Please amend the manuscript to make this distinction clear: if events are free, it can be mentioned early and dismissed as a potential factor influencing attendance. If there is a cost to attendees, then this should be considered as part of the implications of moving online and should be canvassed in the discussion.

The discussion section is limited and based mostly around the ‘digital divide’, however, there are several interesting avenues the authors could discuss their findings in light of. For instance, for the Ireland events in particular it appears there was a focus on in-person events held on university campuses: there is currently no discussion on whether the shift to online may have facilitated a greater breadth of attendees by democratising the venue and being more open to attendees with less formal education who otherwise might feel uncomfortable at a university-based event.

Currently missing from the discussion is any form of recommendation or reassurance for those running such outreach events. It would be useful to comment on such events in the context of these results, especially as some event organisers may choose to operate ‘dual mode’ or provide some online events to help attract a wider audience. It would also be interesting to discuss whether delivering events online was cost-effective to organisers and funders as a way of reaching a broader range of attendees perhaps in addition to in-person events.

Overall, I think the authors haven’t done their study justice in the discussion: their findings raise several interesting points about whether online delivery might attract a wider range of attendees to such events and better achieve goals to connect with audiences with less formal education. At least this should be discussed, potentially in the form of recommendations for event organisers or further research examining events that were forced online in 2020/21. This could also be better tied back to science/research outreach and the goal of connecting with a broader range of audiences.

Reviewer #4: The manuscript is technically sound, well-written, and all data discussed are clearly presented. The statistics need some further explanation to ensure that they have been rigorously tested. As an example, you refer to four 'cases' but it is not clear how these were tested statistically, and how classifying these into 4 cases is important. Overall, the paper demonstrates an important finding, that shifting public science events online can polarise the audience by strengthening attendance from higher educated groups whilst also attracting those with no formal qualification. I felt that the manuscript was rather too simplistic in this analysis however, as there was no discussion about other interesting demographics and how these may be affected by a move to an online delivery. It was also not explained why Malta and Ireland were chosen for this study, and no recommendations were given for improving online events to attract a wider demographic. I would be interested to see a revised version that presents evidence for future event organisers (and other science communication practitioners) to use to ensure greater representation of diverse society at these events.

6. PLOS authors have the option to publish the peer review history of their article (what does this mean?). If published, this will include your full peer review and any attached files.

Reviewer #1: No

Reviewer #2: **Yes: **Katy Thomas

Reviewer #3: No

Reviewer #4: No

---

## [Author Response · Author response to Decision Letter 0]

29 Dec 2021

1. Thank you for including your ethics statement on the online submission form: "This research received ethical approval from an ethics committee at Trinity College Dublin.". To help ensure that the wording of your manuscript is suitable for publication, would you please also add this statement at the beginning of the Methods section of your manuscript file. 

[This line has now been added to the manuscript]

2. Please amend the title either on the online submission form or in your manuscript so that they are identical. 

[The online submission title has been changed to match the manuscript title]

3. Please include a copy of Table 4. 

[Table 4 is now included]

4. We note your current Data Availability statement: "The data underlying the results presented in the study are available from the authors on request." 

[The Data Availability statement has now been changed to say: "The anonymized dataset underlying the results presented in the study will be published on the open-access database Zenodo following acceptance and prior to publication, in line with PLOS ONE data accessibility policy."]

---

## [Editor Report · Decision Letter 1]

6 Jan 2022

How does moving Public Engagement with Research Online Change Audience Diversity? Comparing Inclusion Indicators for 2019 & 2020 European Researchers’ Night events.

PONE-D-21-27097R1

Dear Dr. Roche,

We’re pleased to inform you that your manuscript has been judged scientifically suitable for publication and will be formally accepted for publication once it meets all outstanding technical requirements.

Kind regards,

Dylan A Mordaunt, MB ChB, MPH, MHLM, FRACP, FAIDH

Academic Editor

PLOS ONE

Additional Editor Comments (optional):

Thank you for your resubmission. I accept the responses and this now meets the criteria for publication.
---

## [Editor Report · Acceptance letter]

4 Mar 2022

PONE-D-21-27097R1 

How does moving Public Engagement with Research Online Change Audience Diversity?Comparing Inclusion Indicators for 2019 & 2020 European Researchers’ Night events. 

Dear Dr. Roche:

I'm pleased to inform you that your manuscript has been deemed suitable for publication in PLOS ONE. Congratulations! Your manuscript is now with our production department. 

Kind regards, 

on behalf of

Dr. Dylan A Mordaunt 

Academic Editor

PLOS ONE